# JAX, M.D.
## A Framework for Differentiable Physics

**Samuel S. Schoenholz**
Google Research: Brain Team
schsam@google.com

**Ekin D. Cubuk**
Google Research: Brain Team
cubuk@google.com

## Abstract

We introduce JAX MD, a software package for performing differentiable physics simulations with a focus on molecular dynamics. JAX MD includes a number of physics simulation environments, as well as interaction potentials and neural networks that can be integrated into these environments without writing any additional code. Since the simulations themselves are differentiable functions, entire trajectories can be differentiated to perform meta-optimization. These features are built on primitive operations, such as spatial partitioning, that allow simulations to scale to hundreds-of-thousands of particles on a single GPU. These primitives are flexible enough that they can be used to scale up workloads outside of molecular dynamics. We present several examples that highlight the features of JAX MD including: integration of graph neural networks into traditional simulations, meta-optimization through minimization of particle packings, and a multi-agent flocking simulation. JAX MD is available at www.github.com/google/jax-md.

## 1 Introduction

The past few years have seen an explosion of progress at the intersection of machine learning, automatic differentiation, and the physical sciences [1, 2]. Deep neural network models of quantum mechanical energies have become ever more accurate [3–11], graph networks have been designed to simulate systems by observing their dynamics [12–15], and deep networks have improved upon classical approaches to protein folding [16]. Combining classical simulation environments with deep learning or optimizing them directly via automatic differentiation has led to significant advances including: end-to-end learning of protein structures [17, 18], the inverse design of photonic crystals [19], and structural optimization via reparameterization using convolutional networks [20]. Finally, significant progress in developmental psychology has centered around combining intuitive physics simulations with probabilistic programming [21].

This flurry of excellent research has highlighted a significant source of inefficiency: we lack general purpose simulation environments that can easily be integrated with existing machine learning tools. For example, despite the rapid progress developing neural network models of quantum mechanical systems, it is difficult to incorporate these models into classical simulation environments such as LAMMPS [22], HOOMD-Blue [23, 24], and OpenMM [25]. Currently, integrating machine learning models into existing simulations requires the construction of custom software "bridges" [26–31]. This is a significant hindrance to the adoption of machine learning methods in practice. Moreover, most research into differentiable simulation environments are bespoke pieces of software written for a single application. As such, currently research at the intersection of simulation and machine learning requires practitioners to be experts, not only at machine learning, but also at writing the simulations themselves.

Here we introduce JAX MD, which is a software package that makes it easy to write physics simulations that naturally integrate with machine learning models in the flourishing JAX [32, 33] ecosystem.

At its core, JAX MD provides a collection of lightweight primitive operations that are useful in developing and analyzing a broad range of physics simulations. By composing these primitives, JAX MD also provides standard molecular dynamics simulation environments and interaction potentials. To help researchers analyze simulations, we include tools that can be used visualize particle trajectories within Jupyter notebooks [34]. Finally, we provide several state-of-the-art neural network architectures that can easily be incorporated into simulations no modifications of the simulation code.

We begin with a brief overview of JAX and JAX MD. We then provide benchmarks against classic simulation software before describing the structure of the library. Finally, we discuss several examples that illustrate the use of JAX MD to perform research at the intersection of simulation and machine learning including:

- Training and deployment of a neural network potential in simulation.
- Differentiation through simulation for structural optimization.
- A simulation of flocking behavior.

Each example features an accompanying Colaboratory notebook that reproduces the results and includes more details about the use-case. While these examples are designed to be illustrative, they involve issues faced by researchers in practice. Moreover, each of these examples would be difficult to implement using existing simulation infrastructure.

## 2 Related Work

Automatic differentiation has enjoyed a rich history in the physical sciences [35]; recent applications include: structural optimization [20], quantum chemistry [36], fluid dynamics [37–39], computational finance [40], atmospheric modelling [41, 42], optimal control [43], physics engines [44], protein modelling [17, 18], and quantum circuits [45]. Tools such as DeepChem [46] have consolidated useful best practices for applying deep learning to chemistry. Despite significant work on the topic and a large number of high quality automatic differentiation libraries [47–51, 33, 52], the number of general purpose simulation environments that are integrated with automatic differentiation is scarce.

One exciting recent development in this direction in DiffTaichi [53]. DiffTaichi augments the newly developed Taichi [54] programming language with automatic differentiation capabilities. The Taichi language is a performance oriented language for writing simulations and additional automatic differentiation capabilities offer promising opportunities. There are several contrasts between JAX MD and DiffTaichi. First, JAX MD aims to provide simulation capabilities to a strong machine learning library. Thus, we expect to have easier integration with machine learning models and more flexible automatic differentiation support. Second, JAX MD provides simulation environments that are ready-to-use for research in the physical sciences. However, since Taichi is built from the ground up to deliver fast simulations we expect DiffTaichi to offer a performance advantage relative to JAX MD for many workloads. Nonetheless, as we discuss below, the design of JAX and XLA allows us to produce high-performance code that is competitive with existing software.

## 3 Architecture

We begin by briefly describing JAX before discussing the architectural choices we made in designing JAX MD. JAX is the successor to Autograd and shares key design features. As with Autograd, the main user-facing API of JAX is in one-to-one correspondence with Numpy [55], the ubiquitous numerical computing library for Python. On top of this, JAX includes a number of higher-order function transformations that can be applied to any function written in terms of JAX primitives. Examples of such transformations are: automatic differentiation (`grad`), vectorization on a single device (`vmap`), parallelization across multiple devices (`pmap`), and just-in-time compilation (`jit`). These function transformations are arbitrarily composable, so one can write e.g. `jit(vmap(grad(f)))` to just-in-time compile a function that computes per-example gradients of a function $f$. JAX makes heavy use of the accelerated linear algebra library, XLA, which allows compiled functions to be run with a single call on CPU, GPU, or TPU. This effectively removes execution speed issues that generally face Python programs. Moreover, this just-in-time compilation allows XLA to perform whole-program optimizations that can considerably improve execution speed.

Unlike most physics simulations, JAX MD adopts a functional style similar to JAX with immutable data and first-class functions. In a further departure from most physics software, JAX MD features no classes or object hierarchies and instead uses a data driven approach that involves transforming arrays of data. JAX MD often uses dataclasses to provide minimal organization to collections of arrays. This functional and data-oriented approach complements JAX and makes it easy to apply the range of function transformations that JAX provides. JAX MD makes extensive use of automatic differentiation and automatic vectorization to concisely express ideas (e.g. force as the negative gradient of energy) that are challenging in more conventional libraries. Additionally, we include a number of higher-order functions that make it particularly easy to define custom interaction potentials. Since JAX MD leverages XLA to compile whole simulations to single device calls, it can be entirely written in Python while still being fast enough to perform research. Together this means that simulations implemented in JAX MD tend to be very concise and look like textbook descriptions of the subject.

## 3.1 Benchmarks

While our use of JAX and XLA may provide a more productive research environment than conventional simulation packages, it does have several drawbacks. Most significantly, the primitives exposed by XLA are sometimes at odds with computations that are commonplace in molecular dynamics. Notably, XLA requires that shapes be statically known at compile-time and it is often challenging to use complex data-structures. While JAX MD is fast enough for many research applications, since XLA has been optimized for machine learning workloads, JAX MD is still slower than production quality MD packages using custom CUDA kernels. This is especially true on the CPU backend, where JAX MD currently struggles to scale to large systems. Here we include benchmarks against two of the most common and high-performance molecular dynamics simulation environments: LAMMPS and HOOMD-Blue.

To compare performance, we simulate Lennard-Jones particles using JAX MD, LAMMPS, and HOOMD-Blue. This is a ubiquitous performance benchmark in the molecular dynamics literature. In each case we consider a system of $N$ particles simulated for $10,000$ steps. We include a small $N = 216$ particle system as well as a larger system on each platform. For the larger simulations we used neighbor lists with a cutoff of $3\sigma$ and a halo of $0.5\sigma$. LAMMPS updates the neighbor list every 10 steps whereas both HOOMD-Blue and JAX MD update their neighbor list whenever a particle moves by more than the halo distance.

|  | CPU | | K80 GPU | | TPU | |
|---|---|---|---|---|---|---|
|  | N=216 | N=10k | N=216 | N=64k | N=216 | N=10k |
| JAX MD | 1.9 | 700 | 0.09 | 10.5 | 0.02 | 360 |
| LAMMPS | 0.1 | 120 | 0.20 | 4.0 | - | - |
| HOOMD-BLUE | 0.2 | 120 | 0.07 | 4.5 | - | - |

Table 1: **Comparison of performance between JAX MD, HOOMD Blue, and LAMMPS.** N denotes the number of atoms used in the simulation. Time is reported in units of milliseconds-per-step.

We list the various performance benchmarks in table 1. For small systems on GPU we see that JAX MD performs comparably to the two classic simulation environments. On larger systems, JAX MD incurs approximately a $2.4\times$ slowdown. On CPU, we see that JAX MD is about a factor of 10 slower than either HOOMD-Blue or LAMMPS. On TPU we see that JAX MD performs well for small systems (although a quantitative comparison is difficult, since TPU has limited support for 64-bit arithmetic). However, TPUs perform badly on large systems due to poor neighbor list performance. Future work will focus on improving execution speed on CPU and TPU.

## 3.2 Primitives

We now discuss the structure of JAX MD, starting with primitive operations that are generally useful for constructing simulations based on elements interacting in two- or three-dimensions. This is not meant to be an exhaustive discussion of the features of JAX MD, but is meant to highlight particularly salient points.

### 3.2.1 Spaces

To describe a physical system we must first define the space in which the system lives. In many cases, this space is constructed from subsets of $\mathbb{R}^D$ with $D = 2$ or $D = 3$. For notational convenience and concreteness we will focus on the two-dimensional setting, but we note that extensions to higher dimensions are trivial. In the simplest setting, the space will simply be all of $\mathbb{R}^D$ and these are called "free" boundary conditions. In many cases it is useful to incorporate "periodic boundary conditions". When using periodic boundary conditions, the space is a rectangle $U = [L_x, 0] \times [0, L_y]$ with the association that for any vector $\vec{r}_i \in U$ and $n_x, n_y \in \mathbb{Z}$, $\vec{r}_i = \vec{r}_i + n_x L_x \vec{e}_x + n_y L_y \vec{e}_y$. $U$ together with this identity forms periodic boundary conditions in which elements can "wrap around" the sides of the space. Topologically, this describes a space that is homeomorphic to a two-torus.

In JAX MD we describe spaces by a pair of functions. First, a function $\vec{d} : \mathbb{R}^D \times \mathbb{R}^D \to \mathbb{R}^D$ computes the displacement between two particles. This function can in turn be used to define a metric on the space $d : \mathbb{R}^D \times \mathbb{R}^D \to \mathbb{R}$ by $d(\vec{a}, \vec{b}) = \|\vec{d}(\vec{a}, \vec{b})\|_2$. Note, that in systems with periodic boundary conditions $\vec{d}$ computes the displacement between a particle and the nearest "image" of the second particle. Second, a shift function $\mu : \mathbb{R}^D \times \mathbb{R}^D \to \mathbb{R}^D$ must be defined that moves a particle by a displacement vector. In JAX MD the boundary conditions discussed above are implemented as `d, mu = space.free()` and `d, mu = space.periodic(L)` respectively.

The `displacement` functions are defined for a single pair of particles. However, it is often desirable to compute the displacement for all pairs in a set of $N$ particles, $d_{\text{pair}} : \mathbb{R}^{N \times D} \times \mathbb{R}^{N \times D} \to \mathbb{R}^{N \times N \times D}$, between a set of particles and their neighbors, $d_{\text{nbrs}} : \mathbb{R}^{N \times D} \times \mathbb{R}^{N \times N_{\text{nbrs}} \times D} \to \mathbb{R}^{N \times N_{\text{nbrs}} \times D}$, or between ordered pairs $d_{\text{bond}} : \mathbb{R}^{N \times D} \times \mathbb{R}^{N \times D} \to \mathbb{R}^{N \times D}$. For convenience JAX MD provides several light-weight wrappers around JAX's automatic vectorization that allow us to transform displacement functions for a pair of particles into these respective forms: `d_pair = space.map_pair(d)`, `d_nbrs = space.map_neighbors(d)`, and `d_bond = space.map_bond(d)`.

### 3.2.2 Interactions[1]

A key component of any simulation is the definition of interactions between elements of the simulation. The most common way of defining interactions in physics is via an energy function (similar to a loss for a neural network). Given a collection of $N$ elements, $\vec{r}_i \in \mathbb{R}^D$ with $1 \leq i \leq N$, an energy is a function $U : \mathbb{R}^{N \times D} \to \mathbb{R}$. With the energy in hand one can then perform a wide range of simulations using variants on Newton's laws: $m\ddot{\vec{r}}_i = -\nabla_{\vec{r}_i} U$ where $m$ is the mass. Rather than defining an energy as an interaction between all the elements of the simulation simultaneously, it is common to define a "pairwise" energy based on the displacement between a pair of elements, $u(\vec{r}_i - \vec{r}_j)$. In this case the total energy is then defined by the sum over pairwise interactions, $U = \frac{1}{2} \sum_{i \neq j} u(\vec{r}_i - \vec{r}_j)$.

In JAX MD the energy can be any differentiable function and forces are computed using automatic differentiation. To facilitate the construction of energy functions from pairwise interactions, JAX MD provides several higher-order functions that take functions that act on pairs of elements and transforms them to operate on an entire system. In analogy to the vectorization of spaces, JAX MD provides the functions `U = smap.pair(d, u)`, `U = smap.neighbors(d, u)`, and `U = smap.bond(d, u)` that take a pairwise function and promotes it to act on all pairs, all neighbors, or ordered pairs respectively.

### 3.2.3 Spatial Partitioning

In the analysis and simulation of physical systems, many quantities of interest are local in the sense that they depend only on pairs of elements, $(i, j)$, such that $\|\vec{r}_i - \vec{r}_j\| < \sigma$. In this case, it is wasteful to compute the result for every pair of elements, which scales as $\mathcal{O}(N^2)$, since the number of nonzero contributions often only scales as $O(N)$. To facilitate these computations, JAX MD includes two standard primitive operations to spatially partition systems into neighborhoods of size $\sigma$: first, `partition.cell_list` takes a collection of elements and places them into "cells" where each cell is has size $\sigma$; second, `partition.neighbor_list` converts lists of positions into lists of neighboring elements within $\sigma$. Note that `partition.neighbor_list` uses `partition.cell_list` under the hood to construct the list of neighbors in $\mathcal{O}(N \log N)$ time.

### 3.2.4 Neural Networks

We include several neural network components, built on top of Haiku [56], that make it easy to build neural networks that can be incorporated into simulations. In particular, we provide code to compute popular featurizations that have been used in the literature [57–60]. Using these features we provide an implementation of the Behler-Perrinello architecture [61], which is a popular neural network for approximating quantum mechanical energies. We also provide a number of building blocks to construct graph neural networks based on the GraphNet library [62] along a standard graph network recently used to model glasses [63].

## 3.3 Higher Level Functions

We now briefly describe functions that can be used to perform molecular physics simulations. Here the elements of the simulation will typically be particles or atoms.

### 3.3.1 Standard Dynamics and Simulations

JAX MD supports simple constant energy (NVE) simulation as well as Nose-Hoover [64], Langevin, and Brownian simulations at constant temperature (NVT). JAX MD also supports Fast Inertial Relaxation Engine (Fire) Descent [65] and Gradient Descent to minimize the energy of systems. All simulations in JAX MD follow a pattern that is inspired by JAX's optimizers. In particular, simulations are defined by pairs of two functions: an initialization function and an update function. For example, to construct an NVE simulation one would write, `init_fn, update_fn = simulate.nve(energy_fn, shift_fn, dt=1e-3)`. The initialization function, `state = init_fn(key, positions)`, takes particle positions and returns an initial simulation state; the update function, `state = update_fn(state)`, takes a state and applies a single update step to the state. Keyword arguments fed to the update function get passed to energy functions as well as the displacement and shift functions. This is useful, for example, to simulate a system with a time-varying potential by writing `state = update_fn(state, t=t)`.

### 3.3.2 Standard Energy Functions

We provide a number of energy functions including several pair potentials - Lennard-Jones [66], soft-sphere [67], Buckingham-type [68–70], Gupta [71], and Morse [72] - as well as the Stillinger-Weber [73] three-body potential, the Embedded Atom [74] many-body potential, and soft-spring bonds. For pairwise potentials, we provide three functions: one giving the functional form (e.g. `E = soft_sphere(dr)`), one thin wrapper around `smap.pair` that maps the form to all pairs of particles (e.g. `energy_fn = soft_sphere_pair(d)` where d is a displacement function), and one thin wrapper around `smap.neighbors` that maps the form over all neighbors (e.g. `neighbor_fn, energy_fn = soft_sphere_neighbor_list(d, box_size)`). We also include wrappers around the neural network components to build neural networks specifically designed to predict energies as described in section 4.1. Forces can easily be computed using a helper function `quantity.force(energy_fn)` which is a thin wrapper around `jax.grad`. The automatic computation of derivatives and vectorization make it significantly easier to add new potentials to JAX MD than it is to add potentials to classical MD packages.

## 4 Three Vignettes

### 4.1 Neural Network Potentials[2]

In this section we describe the training and deployment of two state-of-the-art neural networks: the Behler-Parrinello neural network [61] and a graph neural network. We will consider a common task in the field of empirical potentials research: fitting a neural network model to quantum mechanical energies and forces. We will use open source data from a recent study of silicon in different crystalline phases [75]. Here, Density Functional Theory (DFT) - the standard first principles quantum mechanics technique - was used to simulate 64 Silicon atoms at three different temperatures. We follow a standard procedure in the field and uniformly sample these trajectories to create 50k configurations that we split between a training set, a validation set, and a test set.

To create our neural network potential we can write `init_fn, energy_fn = energy.behler_parrinello(d)` or `init_fn, energy_fn = energy.graph_network(d)` which uses Haiku [56] to construct the energy model. The model itself is given by two functions: the initialization function, `params = init_fn(key, input_shape)`, instantiates parameters of the model; the energy function, `E = energy_fn(params, R)`, evaluates the model given parameters and positions. Here the energy function evaluates the energy for a single configuration; however, for training we often want to evaluate the model over an entire batch of data. To do this, we can use JAX's automatic vectorization as `vectorized_energy_fn = vmap(energy_fn, (None, 0))` to vectorize only over the positions.

For the Behler-Parrinello architecture we train for 800 epochs using momentum with learning rate $5 \times 10^{-5}$ and batch size 10. For the GNN we train for 160 epochs using ADAM with a learning rate of $1 \times 10^{-3}$ and batch size 128. We use a loss that is the sum of two terms: the Mean Squared Error (MSE) between the network outputs and the quantum mechanical energies and the MSE between the forces predicted by the model and the quantum mechanical forces. While the Behler-Parrinello network is explicitly invariant to global rotations, the graph neural network is not. Therefore, for the graph neural network we additionally augment the data with random rotations and flips.

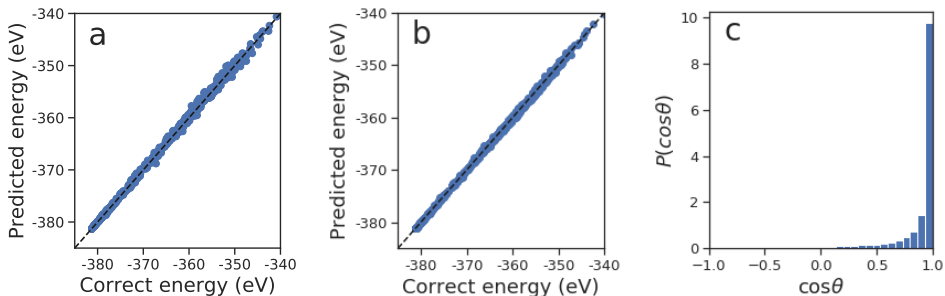

Figure 1: **Empirical potentials trained on DFT.** (a) shows the agreement between predicted energies by a Behler-Parrinello neural network and the correct DFT energies for test samples. (b) shows the agreement between energies predicted by the GNN and the correct DFT energies. (c) the distribution of the per-particle cosine-angle between the correct force and the predicted force for the graph neural network.

We plot the result of training these two networks in fig. 4.1. Fig. 4.1 (a) and (b) show the predicted energy on the test set against the ground truth labels for both network architectures. We find that the graph neural network achieves a root-mean-squared error of 2.8 meV/atom. We can construct forces for these models using JAX MD along with JAX's automatic differentiation as `force_fn = quantity.force(energy_fn)`. We emphasize that the step of computing forces would be extremely laborious in traditional MD packages. In fig. 4.1 (c) we show the cosine-angle between the forces computed from the graph neural network model and the ground-truth forces computed using density functional theory. We observe strong correlations between the predicted forces and the exact forces.

Now that the model is trained, we can easily deploy it into our simulations. To do this, it is useful to "bake in" the parameters using Python's partial evaluation, `energy_fn = functools.partial(energy_fn, params)`. This energy function can now be used as the input to any of the JAX MD simulation environments. To demonstrate this, we simulate a larger system of 64,000 atoms using the graph neural network to compute forces. The graph neural network provides accurate energies and forces for a system that is far larger than could possibly be simulated with DFT. A snapshot of this simulation is shown in fig. 2.

## 4.2 Optimization Through Dynamics[3]

Here we describe how one can differentiate through a simple simulation to optimize the parameters of the simulation. We will use a simple model that is widely studied in condensed matter physics: the packing of spheres into a box. In two-dimensions, when soft disks of equal radius are compressed inside a box they pack regularly into a hexagonal grid. When the spheres have different sizes, however,

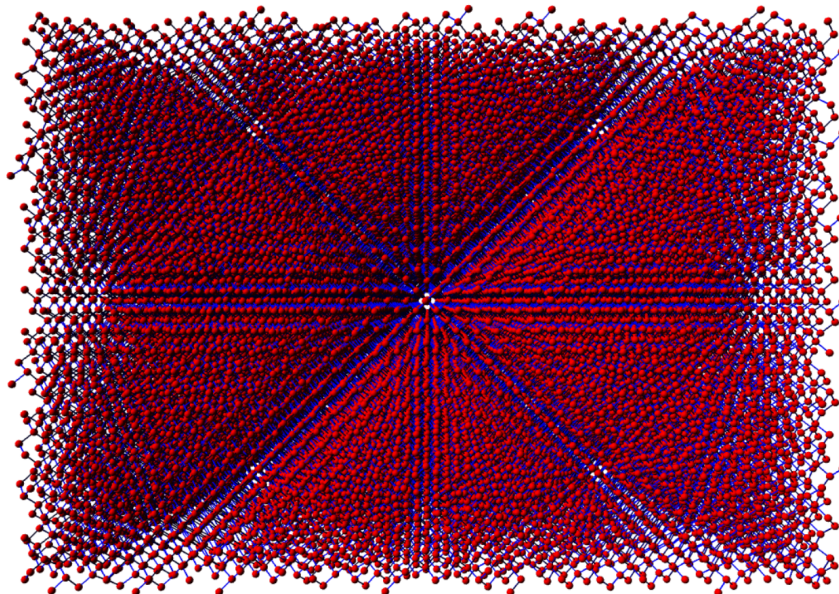

Figure 2: **Simulation using a GNN.** Here we show a snapshot of a 64,000 atom NVT simulation using a graph neural network energy function trained on DFT data.

the hexagonal order can be interrupted and the resulting structure becomes random. These random structures are often referred to as "jammed" [67]. We will consider a system with spheres of two distinct sizes: half of the disks will have diameter 1 and half will have diameter $D$. We will adjust the size of the simulation box so that the fraction of space taken up by the disks remains invariant as we tune $D$.

In fig. 3 (a) we see energy minimizing configurations for several different values of $D$. When $D = 1$ (bottom right) the system has its characteristic hexagonal order and as $D$ shrinks the configuration gets more disordered until it reaches a maximally frustrated state (top right). As the small particles are further shrunk the system becomes more ordered with local hexagonal order among the large particles and the small particles fitting into interstices (top left). Fig. 3 (b) shows the energy of local minimia, averaged over 100 different random samples, as a function of $D$. We see that the energy achieves a maximum where the system is maximally frustrated. We would like to find this point of maximum frustration by differentiating through the simulation directly.

To do this we write a function `final_energy = minimize_energy(diameter, key)` that takes a diameter for the small particles as well as a random key. The function generates a random configuration of particles, minimizes them to their nearest minimum, and then returns the energy of the final configuration. To generate fig. 3 (b) the minimization function can be vectorized separately over diameters and random keys,

```
vectorized_minimize_energy = vmap(vmap(minimize_energy, (None, 0)), (0, None))
```

Since each energy calculation is the result of a differentiable simulation, we can differentiate through the minimization with respect to $D$. This allows us to find extrema of the minimized-energy as a function of diameter using first-order optimization methods. This is implemented in JAX MD as, `dE_dD_fn = grad(minimize_energy)`. Of course the `dE_dD_fn` function can be vectorized in the same way to aggregate statistics from an ensemble over different values of the diameter.

This gradient is plotted in Fig. 3 (c). We see that the gradient is positive and constant for $D < 0.8$ corresponding to the linear increase in the average energy. Moreover, we see that the derivative crosses zero exactly at the maximum average energy. Finally, we observe that the gradient goes back to zero at $D = 1$. This suggests that $D = 0.8$ is the point of maximum disorder, as we found by brute force above. It also shows that $D = 1$ is the minimum energy configuration of the diameter.

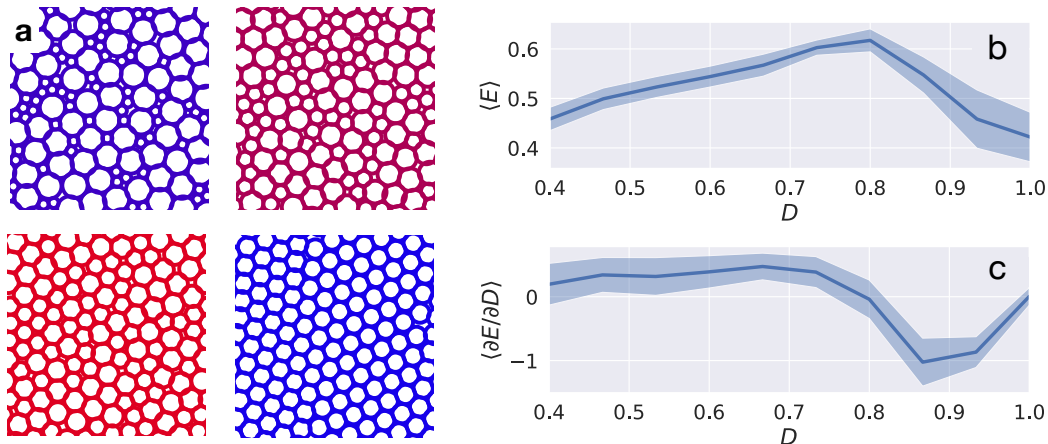

Figure 3: **Finding maximally frustrated states.** (a) Configurations for varying small-particle diameter $D$. (b) The average energy and the standard deviation of the energy at $D$. (c) The derivative we calculate by differentiating through energy optimization by gradient descent as a function of $D$.

Although we hadn't hypothesized it, we realize this must be true since $D < 1$ states are symmetric with $D > 1$ as we keep the total packing fraction constant.

## 4.3    An Energy Model for Flocking[4]

We will now use the primitives provided by JAX MD to build a flocking simulation inspired by Reynolds' landmark paper [76]. This is a simulation that looks very dissimilar from most conventional physics simulations. Nonetheless, the primitives that JAX MD provides allow us to write the simulation in such a way that it scales it to tens-of-thousands of agents (that Reynolds calls "boids"). Reynolds notes that in nature flocks do not seem to have a maximum size, but instead can keep acquiring new boids and grow without bound. He also comments that each boid cannot possibly be keeping track of the entire flock and must, instead, be focusing on its local neighborhood. As such, we will use neighbor lists to keep track of nearby boids and scale the simulation.

Reynolds proposes three simple and local rules that boids might try to follow: alignment causing boids to point in the same direction as their neighbors, avoidance stops boids from colliding with nearby boids, and cohesion that causes boids to point towards the center of mass of their neighbors. We will represent boids by a position, $\vec{r} \in \mathbb{R}^2$, and an angle, $\theta$. We will also include disk-like obstacles along with predators. The obstacles will repel both the boids and the predators. The predators will attempt to move themselves towards nearby boids and the boids will flee from the predators. We also allow the predators to "sprint" every-so-often to chase the boids.

Unlike standard boid simulations, we will treat this as an energy model such that low energy configurations satisfy Reynolds' three properties as well as our own constraints on obstacles and predators. As such, we define two energy functions,

$$E_{\text{boid}} = E_{\text{align}} + E_{\text{avoid}} + E_{\text{cohesion}} + E_{\text{obstacle}} + E_{\text{boid-predator}} \quad (1)$$

$$E_{\text{predator}} = E_{\text{predator-boid}} + E_{\text{obstacle}} \quad (2)$$

The simulation will be defined so that boids and predators move in the direction they are pointing at each step, but have their position and orientation updated to minimize their contribution to the above energy.

Although this simulation is only loosely related to standard physics simulations, we still make use of `space.periodic` to compute displacement and move the boids. We also use neighbor lists so that each boid only has to consider its local neighborhood. A figure showing the behavior of the simulation can be seen in fig. 4. The boids move cohesively and swarm around obstacles while the predators chase them around the simulation. While this simulation can now be used as a standard flocking

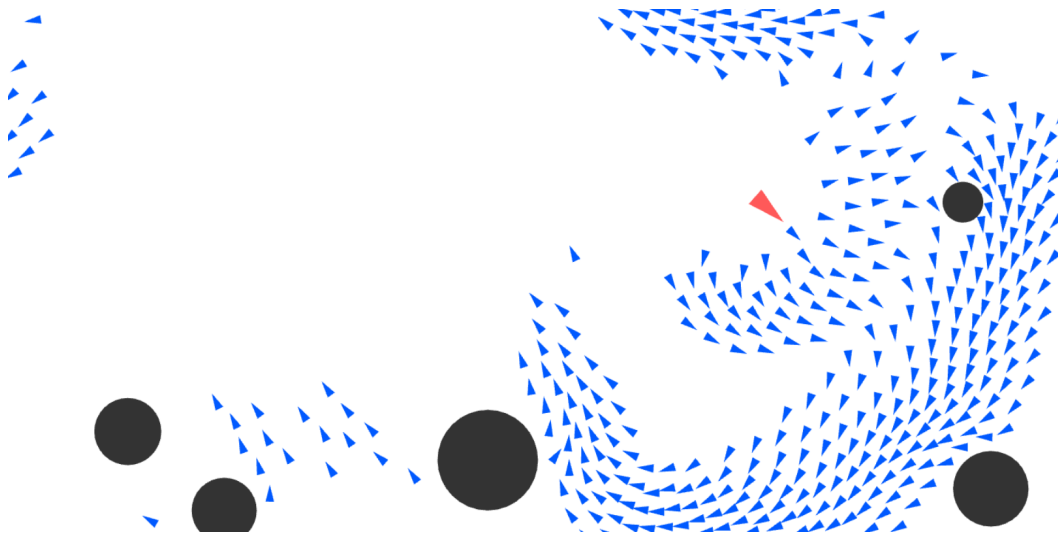

Figure 4: **Flocking simulation.** Blue arrows represent boids, red arrow represents a predator, and black circles are obstacles.

simulation, JAX MD allows us to take it a step further. Since the simulation is differentiable with respect to the parameters that control the agents, we can use gradients to train boids and predators as a two-player game (potentially controlled by neural networks), ensemble populations for efficient evolution, and meta-learn environmental hyperparameters for interesting emergent behavior. This would be an interesting avenue for future experiments.

## 5 Conclusion

We have described recent work on a general purpose differentiable physics package. We have demonstrated several instances where JAX MD can simplify research and accelerate researcher iteration time. We believe that JAX MD will enable new avenues of research combining developments from machine learning with the physical sciences.

## Broader Impact

We believe there is significant potential when deep learning, automatic differentiation, and simulation meet. We are excited about the possibilities of larger and more accurate simulations of quantum mechanical systems. If we can reliably simulate and optimize these systems at scale it has the potential to revolutionize many problems that are currently hard: from material design to drug discovery. Of course, if this program is successful enough to provide actual benefits to benevolent technologies it surely has the potential to be used in an ethically questionable manner to e.g. design weapons. We are also excited about the potential of marrying the extreme expressivity of machine learning models with the excellent generalization capabilities of physical laws to improve our ability to do science.

## Acknowledgments and Disclosure of Funding

We are indebted to early users of JAX MD who provided valuable feedback on early versions of the project, especially Carl Goodrich, Ella King, Michael Brenner, Kareem Hegazy, Ryan Coffee, Rouxia Chen, Matheiu Bauchy, Luke Metz, and Amil Merchant. We would especially like to thank Carl Goodrich for contributions to the library, including the custom potential colaboratory notebook linked in section 3.2.2. We would also like to thank Rouxia Chen for rigorously testing and improving the BKS potential. We thank the entire JAX and XLA teams for making making such outstanding libraries that we were able to build on. In particular, we appreciate useful feedback and assistance from Matthew Johnson, Skye Wanderman-Milne, James Bradbury, and Roy Frostig. We additionally

thank Jascha Sohl-Dickstein for useful feedback along with the Google Brain team for their help and support. Google is the sole source of funding for this work.

## Footnotes

[1]Colaboratory notebook at github.com/google/jax-md/notebooks/custom_potentials.ipynb

[2]Colaboratory notebook at: github.com/google/jax-md/notebooks/neural_networks.ipynb

[3]Colaboratory notebook at: github.com/google/jax-md/notebooks/meta_optimization.ipynb

[4]Colaboratory notebook at: github.com/google/jax-md/notebooks/flocking.ipynb

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
