[Supplementary Material · supplementary.pdf]

**JAX MD: Supplementary Material**

We include anonymized code for JAX MD as well as the various iPython notebooks in the supplementary material. We also have an anonymized version of the code uploaded on github at www.github.com/conference-submitter/jax-md. The paper includes links to iPython notebooks that will execute using Google Colaboratory with a free GPU.