[Reviews · NeurIPS 2020]

Review 1

Summary and Contributions: The paper presents an extension to the JAX high-performance machine learning library, that allows for the integration of multi-body simulations into neural networks research via implementing automatic differentiation for the simulations. The library balances runtime efficiency crucial to physics simulations simulations and ease of implementation to stimulate research via enabling rapid prototyping. It’s achieved by using the accelerated linear algebra library XLA. One of the core contributions seems to be overcoming the limitations of the XLA library stemming from it’s rigidity regarding the fixed number of elements that needs to be known at compile time.

Strengths: Overall, the library seems to be a great tool for a larger number of communities, and I’m particularly excited about getting to know more about its capabilities to integrate physical simulations and GNNs. If NeurIPS is a venue that accepts this type of publications (software frameworks), I think NeurIPS is the perfect venue for a package that is aimed at such a large selection of scientific fields. - Reasonable discussion of alternatives (DiffTaichi, LAMMPS, HOOMD-Blue) - Reasonable discussion of drawbacks of this combination of domain flexibility and efficiency at runtime - Great selection of simulations from different sciences showcased

Weaknesses: - Impact of pairwise energy functions on accuracy not discussed? - Missing discussion on limitations of data-centric representation (i.e. consequences of other choices made but not highlighted, similar to the pairwise energy function above) - Not sure, if NeurIPS accepts papers about frameworks

Correctness: Yes, modest claims supported by evaluation.

Clarity: A bit too fluid, and less structured (the second half is "just a bunch of examples"), but it's difficult to judge, as one doesn't see framework papers that often. Limitations and future work could be more collected at the end, rather than mentioning it throughout the paper.

Relation to Prior Work: Yes.

Reproducibility: Yes

Additional Feedback: Keep going!


Review 2

Summary and Contributions: This paper describes "JAX MD" a software package for performing differentiable physics simulations, with an emphasis on molecular dynamics.

Strengths: In general, this work presents a differentiable physics simulator, inserting itself in an area that has recently been the subject of interest and many works. This differentiation capability obviously allows for the combination of physics models and gradient-based machine learning approaches, such as deep learning models, which is also of interest to many subfields in machine learning. One of the specific strengths of the proposed framework is its usage of JAX syntax, which is in direct correspondence to Numpy syntax, making it simple to learn and use both for creating new and converting existing numerical simulations. In addition to this, its arbitrarily composable primitives that simplify controling important operations such as differentiation, parallelization/vectorization, and compilation also contribute to its strengths.

Weaknesses: As the paper itself points out, due to intrinsic characteristics of the framework, it suffers from some drawbacks in comparison to similar (though non-differentiable) high-performance frameworks, especially, for example, when running large simulations on the CPU.

Correctness: I do not see any errors in this work.

Clarity: The paper is well written and clear.

Relation to Prior Work: The paper discusses its relations to similar frameworks, even being honest about presenting its comparative shortcomings in certain scenarios.

Reproducibility: Yes

Additional Feedback: I am not used to reviewing software frameworks in conference reviewing processes such as these, so I have assigned my evaluation a lower confidence score. Nevertheless, it seems to me, due to the reasons described in the strengths section, that JAX MD will be of significant interest to the machine learning community. It is also an important addition to the JAX ecosystem, both furthering and gaining from the interest that this automatic differentiation framework has recently generated. ======================= Comments after the authors response: After reading the authors' response and the other reviews, I have decided to maintain my previous evaluation. Many of the answers provided by the authors where satisfactory. Thus, I still believe that, while the paper might have some flaws (like some pointed out by R3), it is nonetheless worthy of acceptance. I urge the authors to improve on the points mentioned in the reviews and the rebuttal for the final version.


Review 3

Summary and Contributions: The paper introduces a differentiable physics simulation code base that the authors call JAX MD. The package is based on JAX, a recently very popular automatic differentiation and vectorization library for Python. The submitted paper does not explain what “MD” stands for, but it probably refers to “molecular dynamics”. What is presented is essentially a differentiable physics engine (in particular a particle-based Newtonian physics engine) that can be used for various tasks involving particle-based simulations and differentiation. The paper provides experimental results in optimizing particle interaction potentials represented as neural networks, gradient-based optimization of physics model parameters, and an energy model simulating flocking behavior in swarms. - After author response: I have read the author response. Given the current manuscript and author response, my evaluation (an ok submission, but not good enough, reject) remains the same. The authors have clearly disagreed with me on several points in their response. We are on the same page about disagreeing. The main issues are: (1) the submitted manuscript and the author response give me the impression that the authors see this paper simply as a pointer (advertisement) to their molecular dynamics library and see NeurIPS as a marketing venue more than a venue with traditional research proceedings (which may be fine, as my task is to review the paper and not the authors' intentions); (2) content-wise the submitted manuscript looks like an early draft that needs significant revision and I'm not convinced that the authors agree with this or will address this. Extra feedback: I think the purpose the authors have in mind (mainly marketing their library to the community) would be served by the library's website, online materials, notebooks, etc. In contrast, I believe that research papers are expected to communicate methods, assumptions, facts, experiments, etc. in clear and precise language free from vague marketing language, compatible with existing conventions in paper writing, in a self-contained and reproducible way. Saying "we have included links to three detailed examples that can be run interactively in colab, did the referee notice the colab notebooks?" is not good enough. I think the paper needs to stand on its own, but at this point it reads like a first draft that needs to go through a couple of revision cycles.

Strengths: The presented software library seems to be well-engineered, and I am sure that it can be adopted by the ML and physics communities and lead to potentially impactful research. It is a really respectable and important work to provide well-designed software tools that take advantage of the recent advances in ML frameworks.

Weaknesses: The submitted manuscript does not currently fulfill the standards I would expect from a NeurIPS paper. The main weaknesses are the lack of technical depth (both in the description of the work and the results) and the paper content not being in good shape in terms of language used and the writing style. In my assessment the paper reads like a workshop paper at best. Some reviewers would probably add “lack of novelty” to this list, but I will not list this as a “weakness” as I think there is great value in doing the type of work presented in this paper and providing new tools to the community, even when there are no significant methodological advances. The presented library has runtime performance which is not impressive compared to baselines (as admitted by the authors), and this prevents “performance” from being a strength of this paper. One of my serious problems with this paper is the vague and unscientific “marketing” language used overall, using a lot of very vague qualifiers, unsubstantiated claims, and in general overselling of what has been achieved. This language is not necessary and it really distracts from the value of the research. I would like to strongly encourage the authors to remove this language and stick to research facts and well-defined statements. Some examples of this problematic language are below. Abstract “a fully-differentiable software package”: what is “fully differentiable”? How can you define it? Abstract: “JAX MD includes high-quality implementations”: what is a “high-quality implementation”? Abstract: “state-of-the-art networks that can be seamlessly integrated into these simulations”: What does “seamlessly integrating” mean? What is the alternative when it is not “seamless2? Abstract: “built on a powerful collection of primitives”. What is “powerful”? How do you even define this? Abstract: “make it easy to write exotic, custom, simulations”. What is an “exotic” simulation? Abstract: “easy integration of graph NNs”. What do you exactly mean by “easy integration”? What makes it easy and what would make it “difficult”? Introduction “can be incorporated into these simulations with no additional effort” What do you mean exactly by “no additional effort”? What would constitute an “additional effort” for you? Section 3: “Jac MD leverages …, it can be entirely written in Python while still being extremely fast”. What do you mean exactly by “extremely fast”? And your benchmark results show that the library is indeed not “extremely fast” at all. Section 3: “looks almost verbatim like textbook descriptions of the subject”. What exactly do you mean by “almost verbatim textbook description”. This is extremely vague. You should be showing some example in the paper to make your point. Section 3: “Behler-Perrinello architecture, which is perhaps the most popular NN for computing approximations”. This language is extremely vague. Why don’t you just say “which is a popular”, if you are not sure about it being “the most popular”? Section 3.1: “this is, perhaps, the most standard performance benchmark in MD literature”. Is it or is it not the “most standard” benchmark? Section 4.3: "although this simulation is only loosely related to standard physics simulations". What is a "standard physics simulation" for you? Also "loosely related" is again a very vague thing to say. Conclusion: “several instances where JAX MD can simplify research and accelerate researcher iteration time”. What exactly do you mean by “simplify research” and “accelerate researcher iteration time”. These sentences are very vague, without any evidence, and conversational.

Correctness: The paper makes claims of superior or “extremely fast” performance about the library introduced. However the benchmark results do not support this claim, and this is admitted by the authors in the benchmark section. I think these performance claims were a motivation of the work that has not been realized in practice so far. Until this “extremely fast” performance happens in practice, I would encourage the authors to just revise the paper and homogenize the way they are talking about the performance of JAX MD.

Clarity: The paper is not well written. (See my comments in the "weaknesses".) Please introduce what “MD” stands for, clearly in the introduction. Please, in the introduction, provide a clear explanation of exactly what type of physics simulation your system is targeting. I think what you are doing is not a “general purpose simulation environment” for performing “broad range of physics simulations” as you vaguely state in the paper. It looks like your method is limited to the setting of particle-based Newtonian simulation. Physics simulation covers a very wide spectrum of systems and scales, ranging from particle physics (Standard Model interactions) to cosmology. The type of simulation environment you provide seems to be applicable to settings limited to (or comparable with) molecular dynamics. Please very clearly define the scope and type of simulations that JAX MD is designed for. The first mention of “molecular dynamics” in the paper (that I think what JAX “MD” refers to) shows up in Section 3.1, benchmarks. This is too late. Table 1: are these results in 2d or 3d simulation? Figure 3 is really strange and it takes up a lot of space without really providing any useful information. You could make it smaller and actually provide the code or pseudocode of the corresponding simulation in JAX MD so that the reader can appreciate the way these simulators are constructed in your system. In Section 4.3 in general I don’t see any results being presented and it just ends abruptly without and results. Minor: page "D=2 or D=3... we note that extensions to higher dimensions are trivial" Does it even make sense to have an MD simulation with D>3? What would that space that the particles move in correspond to? Minor: On page 6, the references "Fig 4.1" should be to Fig 1, I think Minor: on page 7 "gradient is positive and constant for D <0.8". Obviously the gradient is not constant D < 0.8, see the figure. Page 7: "this allows to find extrema of the minimized energy". Can you clarify what it means to find the extrema of an already minimized quantity? I think there is a grammar issue here. Figure 2: what are the D values that gave rise to the four configurations on the left hand side. It would be good to provide them.

Relation to Prior Work: The related works section’s first paragraphs reads like an introduction and could actually help improving your introduction section, if you move it there. In the rest of the related works section, you just have 1 (one!) related work cited which is DiffTaichi (Hu et al 2019). There are tons of related works that can be cited, one pointer I can provide is the computational fluid dynamics simulations that are a very big application domain of automatic differentiation outside machine learning.

Reproducibility: Yes

Additional Feedback: I would suggest the authors submit this paper to a venue specialized in ML software packages, such as JMLR Machine Learning Open Source Software track, instead of NeurIPS.


Review 4

Summary and Contributions: The authors introduce Jax MD, a versatile molecular dynamics simulation environment. This is a fully differentiable software package which offers a user-friendly interface to machine learning interaction potentials and to well known classical force fields. This highly optimized package comes to fill out a void in the of ML physical simulations.

Strengths: As a practitioner, I can see myself using this code to do research. It is easy to use as well as having an intuitive interface that makes it easy to build on top.

Weaknesses: As the authors mention, there are certain limitations regarding deploying it on TPUs or by directly competing with more established MD codes (e.g. LAMMPS).

Correctness: Everything in place.

Clarity: It is a nicely written article.

Relation to Prior Work: A simple but informative section is presented.

Reproducibility: Yes

Additional Feedback: An example of a time dependent force field would be very interesting to add to the portfolio of applications of the paper.

[Author Response · NeurIPS 2020]

We would like to begin by thanking the reviewers for their time, their careful reading of our submission, and for the valuable feedback they have provided. We are particularly excited that all four reviewers agreed that the most important aspect of the work – the codebase itself – seemed like a valuable contribution to the *NeurIPS community*: R1: "Overall, the library seems to be a great tool for a larger number of communities, and I'm particularly excited about getting to know more about its capabilities to integrate physical simulations and GNNs.", R2: "...inserting itself in an area that has recently been the subject of interest and many works. This differentiation capability... is also of interest to many subfields in machine learning.", R3: "The presented software library seems to be well-engineered, and I am sure that it can be adopted by the ML and physics communities and lead to potentially impactful research.", and R4: "As a practitioner, I can see myself using this code to do research. It is easy to use as well as having an intuitive interface that makes it easy to build on top."

**Suitability of Venue.** R1: "Not sure, if NeurIPS accepts papers about frameworks" and R3: "I would suggest the authors submit this paper to a venue specialized in ML software packages..." While we agree that software packages are less common submissions, the NeurIPS call-for-papers includes "3. Data, Competitions, Implementations, and *Software*: Benchmarks; Competitions or Challenges; Data Sets or Data Repositories; *Software Toolkits*."

**Performance.** All of the reviewers commented that our performance lags behind traditional MD software; e.g. R2: "...it suffers from some drawbacks in comparison to similar (though non-differentiable) high-performance frameworks, especially, for example, when running large simulations on the CPU." and R4: "As the authors mention, there are certain limitations regarding deploying it on TPUs or by directly competing with more established MD codes (e.g. LAMMPS)." As we mention in the paper, we agree wholeheartedly that there is a lot of room for improvement, especially on CPU and TPU. We are actively working to improve performance across the board. We agree with R3 that our messaging is inconsistent (R3: "...revise the paper and homogenize the way they are talking about the performance of JAX MD."). We will change the language to say that JAX MD is "fast enough to effectively do research with." Since JAX MD's GPU performance is significantly faster than LAMMPS / HOOMD Blue CPU, we feel this is an accurate phrasing.

**Citing prior work.** R3: "The related works section's first paragraphs reads like an introduction and could actually help improving your introduction section, if you move it there. In the rest of the related works section, you just have 1 (one!) related work cited which is DiffTaichi (Hu et al 2019)..." We disagree on this point. We discuss a range of related work in the first paragraph, including a number of fluid dynamics papers dating back to 2005. We wanted to spend more time discussing DiffTaichi since, as far as we are aware, it is the most closely related software to our own. If there are citations that we have missed, we are more than happy to add them.

**Writing.** R3's main issue with the paper was R3: "The main weaknesses are the lack of technical depth (both in the description of the work and the results) and the paper content not being in good shape in terms of language used and the writing style. In my assessment the paper reads like a workshop paper at best." Although phrased differently, R1 seems to share a similar concern, R1: "A bit too fluid, and less structured (the second half is "just a bunch of examples")".

Our goal in writing the paper was to be accessible and interesting to the broad and interdisciplinary audience of NeurIPS. Indeed, we received feedback that a previous version of the paper was overly technical. However, we definitely do not want to come across as "marketing" the work and we appreciate the reviewer for pointing out that it came across this way. We are still working to strike a balance; since most of the reviewers seemed to view the exposition favorably it seems that the writing needs to be tweaked rather than overhauled. We agree with R3's specific suggestions here and we will make these changes to the manuscript to sharpen the language. We also agree with R3 that we should state what MD stands for at the start of the paper and be more precise about the range of physics simulations we consider.

One particular point on writing style: in cases where JAX MD was an improvement over previous MD packages we intentionally refrained from making a juxtaposition (e.g. questions like R3: "What does "seamlessly integrating" mean? What is the alternative when it is not "seamless"?"). We have strong responses to these questions (e.g. previously people wrote bridges between ML software, like TF, and the C / CUDA internals of existing MD systems; here features still need to be differentiated *by hand*), but we thought the *average* NeurIPS reader would get more benefit from learning about what they can do with JAX MD than from a comparison with software packages that they may not have heard of before. R3, do you think the paper would be stronger if we make these comparisons? In either case we can make our use of language more precise.

**Technical Depth.** Above, R3 suggested that the paper lacked "technical depth". Here we disagree with the reviewer. The paper's contribution is principally the software library, which we feel offers ample technical depth for a NeurIPS paper. Moreover, we have included three detailed examples that can be run interactively in colab. While these examples are pedagogical, they are very close to current research (e.g. the GNN we train was used in several papers published this year). Perhaps the structure of the paper obscured the technical depth and we would appreciate feedback. Did the referee notice the colab notebooks or should we emphasize them more? If you did notice them, did you not find the style of conveying content effective?

[Meta-Review · NeurIPS 2020]

All reviewers agree that this paper describes a technically impressive contribution to machine learning research. Although the field of "physics and ML" is relatively small, it is growing, and the techniques underpinning this work offer clear learnings for the wider ML community. Therefore, it appears clear that the audience for this paper is the NeurIPS audience, and the level of contribution is at the level of NeurIPS papers. All reviewers, and R3 in particular had some questions about the style of presentation, and I am content with the rebuttal's statement that the authors will take the feedback on style for a final version.